# In-Silico Evaluation of Glucose Regulation Using Policy Gradient Reinforcement Learning for Patients with Type 1 Diabetes Mellitus

**Jonas Nordhaug Myhre [1,*,†], Miguel Tejedor [2,†], Ilkka Kalervo Launonen [3], Anas El Fathi [4]** and **Fred Godtliebsen [5]**

1   Department of Physics and Technology, UiT-The Arctic University of Norway, 9019 Tromso, Norway
2   Department of Computer Science, UiT-The Arctic University of Norway, 9019 Tromso, Norway; miguel.tejedor@uit.no
3   Department of Clinical Research, The University Hospital of North-Norway, 9019 Tromso, Norway; ilkka.launonen@unn.no
4   The McGill Artificial Pancreas Lab, McGill University, Montreal, QC H3A 2B4, Canada; anas.elfathi@mail.mcgill.ca
5   Department of Mathematics and Statistics, UiT-The Arctic University of Norway, 9019 Tromso, Norway; fred.godtliebsen@uit.no
\*   Correspondence: jonas.n.myhre@uit.no
†   These authors contributed equally to this work.

**Abstract:**  In this paper, we test and evaluate policy gradient reinforcement learning for automated blood glucose control in patients with Type 1 Diabetes Mellitus. Recent research has shown that reinforcement learning is a promising approach to accommodate the need for individualized blood glucose level control algorithms. The motivation for using policy gradient algorithms comes from the fact that adaptively administering insulin is an inherently continuous task. Policy gradient algorithms are known to be superior in continuous high-dimensional control tasks. Previously, most of the approaches for automated blood glucose control using reinforcement learning has used a finite set of actions. We use the Trust-Region Policy Optimization algorithm in this work. It represents the state of the art for deep policy gradient algorithms. The experiments are carried out in-silico using the Hovorka model, and stochastic behavior is modeled through simulated carbohydrate counting errors to illustrate the full potential of the framework. Furthermore, we use a model-free approach where no prior information about the patient is given to the algorithm. Our experiments show that the reinforcement learning agent is able to compete with and sometimes outperform state-of-the-art model predictive control in blood glucose regulation.

**Keywords:**   reinforcement learning; Type 1 Diabetes Mellitus; policy gradient; deep learning; artificial pancreas

## 1. Introduction

Type 1 Diabetes Mellitus (T1DM) is a metabolic disease caused by the autoimmune destruction of insulin-producing beta cells in the pancreas [1]. The role of insulin is to utilize and transport glucose [2]. T1DM patients need life-long external insulin therapy to regulate their blood glucose concentrations. Without insulin, T1DM patients suffer from chronic high blood glucose levels (hyperglycemia) and, conversely, too much insulin causes hazardous low blood glucose levels (hypoglycemia). In fact, fear of hypoglycemia is a major limiting factor of glucose regulation in T1DM [3].

Treatment of T1DM mainly consists of either multiple daily insulin injections (MDI), or through a pump providing a continuous insulin infusion (CSII) [4]. MDI therapy consists of a basal-bolus insulin regimen, where patients take a basal long-acting insulin dose approximately once a day to regulate fasting blood glucose levels, and short-acting insulin boluses around mealtimes to quickly reduce the impact of carbohydrate intake. Bolus insulin is also used for minor adjustments during the day when the blood glucose level is too high. CSII treatment is a different strategy where the patient instead has an insulin pump that continuously infuses insulin. The pump delivers both basal and bolus doses, where the basal rate consists of regularly infused short-acting insulin doses, while the boluses are activated by the user together with meal intakes and to account for hyperglycemia. In both cases, the insulin is administered subcutaneously, i.e., in the fatty tissue just below the skin. In combination with this, the blood sugar levels have to be monitored. This is either done several times per day via manual finger-prick measurements, or via a continuous glucose monitor (CGM) embedded in the subcutaneous tissue [5]. Finally, in collaboration with a physician, the T1DM patient will design a treatment plan based on their individual needs and self-administer insulin according to the plan and self-measured blood glucose concentrations. The goal of the insulin treatment strategy is to keep the blood glucose levels within the normoglycemic range between 70 and 180 mg/dL [6,7].

Due to the demands of everyday life and the fact that patients to a large degree are responsible for treating themselves, the decisions related to the insulin treatment are thus based partly on hard calculations, personal and medical experience, rules of thumb, and, in some cases, just pure guesswork. Although this results in effective treatment when done correctly, it is extremely time-consuming and a constant burden for the patients.

With the improvement of modern treatment equipment, the combination of an insulin pump and CGM invites the addition of a third element, namely a control algorithm to substitute the operation of beta cells in the healthy pancreas. These three elements constitute the artificial pancreas [8,9]. A pump delivers the insulin subcutaneously, which causes delay in the insulin's action compared to normal insulin secretion where the pancreas releases it to the liver via the portal vein. A simple reactive controller based on momentary blood glucose change cannot thus keep up with the delay to avoid high glucose levels after meals. There exists also a delay associated to the subcutaneous blood glucose measurements from the CGM. Besides the insulin action and CGM delays, there are also dynamic factors that cause variation in the patient-specific parameters and complicate the automation of the control process. The effect of exercise on the blood glucose and insulin dynamics is particularly difficult to model and it is a major source of hypoglycemia [10]. .

The only commercial available artificial pancreas system, the Medtronic 670G [11], as well as several do-it-yourself systems, see e.g., [12] and academic systems, e.g., [13] are all hybrid closed loop systems. A hybrid system means that the patient has to provide information to the system about the number of carbohydrates ingested during a meal. A bolus can then be provided, either automatically by the system or by the patient itself based on the estimated carbohydrate amount. This setup is highly prone to errors due to the difficulties of carbohydrate counting in everyday situations [14]. This difficulty is well established in the scientific literature, where the true effect of these errors is still a topic of debate. Among others, Deeb et al. [15] report that carbohydrate-counting errors are not correlated with meal size, while Vasiloglou et al. [16] found that larger meals led to larger estimation errors. On the other hand, Kawamura et al. [17] found that meals with small amounts of carbohydrate tended to be overestimated. Finally, Reiterer et al. [14] note that random errors, such as faulty carb-counting, as opposed to systematic bias errors, are more detrimental to glycemic control. Under- and over-bolusing due to these difficulties presents a significant risk of postprandial hyperglycemia and hypoglycemia. The current strategy to compensate for the counting errors is to let the artificial pancreas temporarily change the basal insulin rate. Despite these issues, the artificial pancreas is currently the most promising option for persons struggling with T1DM with multiple studies showing promising results, both clinical and in-silico [12,18–20].

There are currently two dominant artificial pancreas controller algorithm paradigms, proportional-integral-derivative (PID) control, [11,21], and MPC [22,23]. Model predictive control, in particular, uses a dynamic model with patient-specific parameters to predict the blood glucose curve into the future, where the prediction window is typically four hours, after which the fast-acting insulin's effect has mostly subsided [24]. Afterwards, if the predicted blood glucose curve and its final value is off the glucose target, MPC calculates an optimized sequence of basal rate actions on the model to correct the prediction towards the target while avoiding hypoglycemia. The first action of this sequence is then picked to change the basal rate momentarily, and the whole process is repeated after a while, usually every five or 10 min. The MPC approaches require a good model of the dynamics. In the artificial pancreas system, MPC algorithms are based on glucose-insulin regulatory models that are not able to capture external perturbations, so these algorithms are limited to compensate for the incomplete model used in the artificial pancreas application [25].

In addition to PID control and MPC, there have been investigations into fuzzy logic [26], and more recently techniques from machine learning and statistics, such as Aiello et al. [27], who proposed a blood glucose forecasting approach based on reccurent neural networks. Similarly, Li et al. [28] created a deep learning based forecasting framework based on convolutional neural networks. The control algorithm used in the artificial pancreas system has to learn models that are rich enough and adapt to the system as a whole [25]. Particularly, reinforcement learning (RL), a branch of machine learning that is based on interactive learning from an unknown environment [29] has, in recent years, gained increased attention in artificial pancreas research [30–39]. A complete systematic review of reinforcement learning application in diabetes blood glucose control can be found in [40]. Outside of diabetes-related research, it has been particularly successful in achieving performance that exceeds the level of top human players in strategy games. The examples range from Backgammon in the early 1990's and more recently in the game of Go in 2015, where RL was combined with deep neural networks and Monte Carlo tree search [41,42]. RL allows us to introduce model-free and data driven algorithms that can enable another level of patient individualization [25]. Finally, previous works from the authors have shown promise for the use of RL in the artificial pancreas [32]. In that work, the amount of infused insulin was selected from a fixed and finite list of values, while the blood sugar level was treated as a continuous variable. In addition, there are several recent works using similar methodology [30,33,34,36–39].

In this work we extend the evaluation of RL algorithms for the artificial pancreas and study the performance of Policy Gradient RL algorithms. It is well known in RL literature that policy gradient algorithms are the most suitable for problems where the action space is continuous. This is an important step in the intersection between the RL and diabetes research. Furthermore, we focus on deep Policy Gradient methods due to the flexibility, power and availability of modern neural network approaches [43–46].

We perform in-silico experiments while using the Hovorka model [22] and the trust-region policy optimization of Schulman et al. [45]. Our experiments demonstrate that RL can adapt to carbohydrate counting errors and that RL is flexible enough to treat a population of 100 patients using a single set of training hyperparameters. We consider MPC to be the current state-of-the-art approach and, thus, we compare the performance of the RL agents to that of MPC. Performance is measured through time-in-range (time spent on healthy blood glucose levels), time in hypo-/hyperglycemia, as well as blood glucose level plots for visual inspection.

## 1.1. Related Work

We include a quick overview over the most recent developments in deep reinforcement learning and the artificial pancreas. Particularly, Sun et al. [35] used reinforcement learning to learn the parameters of the insulin pump, specifically the insulin to carb-ratio, and not the insulin action itself. They do not use neural networks in the process. Zhu et al. [38] is quite similar to our work; however, they use PPO, a simpler version of TRPO, and they use the blood glucose level, bg rate, and an

estimate of insulin-on board in the state space. The main difference between their work and this work is that they design a reward function that mimics the natural behaviour of the missing beta-cells, whereas our work focuses on a reward that encodes a more direct approach towards well-established performance measures for T1D therapy (time-in-range, etc.). Finally, Lee et al. [39] proposed a Q-learning approach, where a discrete number of actions modify the given basal rate. They also operate in a dual-hormone approach, where the infusion of glucagon is one of the actions. Their reward function however is quite similar to ours. Finally, they provide an alternative approach to training, where a population level policy is first introduced, followed by individual adaptation to each in-silico patient.

### 1.2. Structure of Paper

We begin with a short introduction to RL in Section 2 followed by a section about in-silico simulation for T1DM in Section 3. In Section 4 we present results and discussions. Section 5 provides concluding remarks and directions of possible future work.

## 2. Theoretical Background

In this section, we present the relevant theoretical background. We start with an introduction to RL, followed by a short section on MPC.

### 2.1. Reinforcement Learning

Informally, RL concerns the behavior of a decision-making agent interacting with its unknown environment. In this framework, the goal is to train an agent to take actions that result in preferable states. Figure 1 shows the agent-environment interaction, where at each time step the agent observes the current state of the environment and performs an action based on that state. As a consequence of this action, the environment transitions to a new state. In the next time step, the agent will receive a positive or negative reward from the environment due to the previous action taken [29].

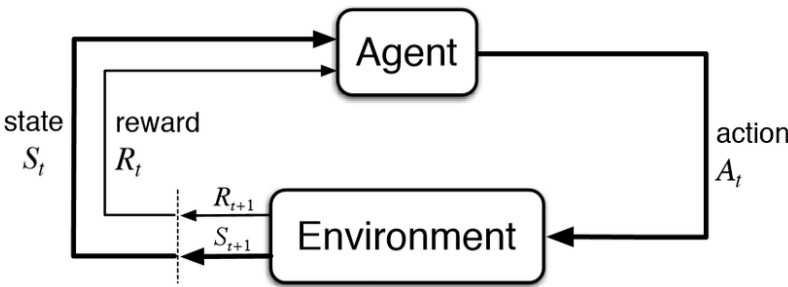

**Figure 1.** The reinforcement learning framework.

The mathematical basis of reinforcement learning is the Markov decision process, which is represented by the tuple $(\mathcal{S}, \mathcal{A}, \mathcal{P}, \mathcal{R}, \gamma)$. $\mathcal{S}$ and $\mathcal{A}$ are the state and the action spaces, respectively, $\mathcal{P}$ contains the state transition probabilities $p(s'|s, a)$ and represents the transition to state $s'$ from $s$ using action $a$. $\mathcal{R}$ contains the rewards, represented by the reward function $r(s, a, s')$, which defines the goal of the problem, and $0 < \gamma \leq 1$ is a discount factor. The mapping from state to action is called the policy, which can be either a deterministic function $\pi : \mathcal{S} \rightarrow \mathcal{A}$ or a set of conditional distributions $\pi(a|s)$, depending on the environment the agent is interacting with. The goal of any RL algorithm is to learn an optimal policy $\pi^*$ that maximizes the expected return it receives over time, which is the accumulated reward over time $G_t = \sum_{k=0}^{\infty} \gamma^k R_{t+k+1}$, where $R_t = r(s_t, a_t, s_{t+1})$. The expected return assuming that the agent starts from the state $s$ and thereafter follows the policy $\pi$

is called the value function $v_\pi(s)$. Concretely, the value function specifies the long-term desirability of states, indicating the total amount of reward that is expected by the agent:

$$v_\pi(s) = E_\pi[G_t|S_t = s] = E_\pi[\sum_{k=0}^{\infty} \gamma^k R_{t+k+1}|S_t = s].$$

Similarly, the expected return assuming that the agent starts from the state $s$, takes action $a$, and thereafter follows the policy $\pi$ is called the *action-value function* $q_\pi(s, a)$:

$$q_\pi(s, a) = E_\pi[G_t|S_t = s, A_t = a] \tag{1}$$

$$= E_\pi[\sum_{k=0}^{\infty} \gamma^k R_{t+k+1}|S_t = s, A_t = a]. \tag{2}$$

The ultimate goal of RL is to find an optimal policy, a policy that is better than or equal to all other policies based on the values of the states. Realizing this goal in practice has led to two different main branches of RL algorithms, value based algorithms and policy based algorithms, see e.g., [47].

Value based algorithms aim to estimate the value of each state the agent observes. Decisions are then made such that the agent spends as much time as possible in valuable states. A policy in value based RL is often simply a greedy search over each action in the given state, where the action that gives the highest value is chosen. In the case of a RL agent controlling e.g., an insulin pump in the T1DM case, such states could be safe blood glucose levels, while states with lower value would be either high or low blood glucose values.

Policy based algorithms change the viewpoint from looking at how valuable each separate state is, to evaluating how good the policy itself is. Given some parametric policy, a performance measure for the policy is defined—most commonly how much reward the agent can get over a certain amount of time. This measure is then optimized using gradient-based methods. For the T1DM case, this performance measure could for example be time-in-range.

### 2.2. Policy Gradient Methods

Policy gradient algorithms consider a parametric policy, $\pi(a|s, \theta) = P(a|s, \theta)$, and the goal is to optimize this policy using gradient ascent with a given performance measure $J(\theta)$ with parameter updates $\theta_{t+1} = \theta_t + \alpha \nabla J(\theta_t)$ [29]. The most common choice for the performance measure is the expected return of the initial state $s_0$, given as

$$J(\theta) = v_\pi(s_0) = \mathbb{E}_\pi \left[ R_0 + \gamma R_1 + \gamma^2 R_2 + \cdots \right].$$

This is equivalent to optimize the value of the initial state—a policy is thus considered to be good if it can generate a lot of reward during the course of an episode.

There are multiple benefits of using policy gradient algorithms; they can be applied directly on continuous action spaces, the policy gradient theorem, introduced below, shows that any differentiable parametric policy can be used and, in the limit deterministic policies, can be modeled by policy gradients, which is useful if we do not want stochastic actions in an online setting—such as in the diabetes case.

One of the key points of policy gradient algorithms is the policy gradient theorem [48]:

$$\nabla J(\theta) \propto \sum_s \mu(s) \sum_a q_\pi(s, a) \nabla \pi(a|s, \theta). \tag{3}$$

where the distribution $\mu$ is the stationary distribution of the states succeeding $s_0$ when following $\pi$. This theorem states that the gradient of the performance measure is proportional to the gradient of the policy itself. This is of great benefit, as it allows the use of any differentiable policy parameterization. The policy gradient theorem allows, with some simple modifications to Equation (3), the formulation

of a simple sample-based algorithm, called REINFORCE. Instead of updating based on summing over all actions, the policy gradient is rewritten using a single sample $S_t$, $A_t$, and the gradient update rule becomes

$$\theta_{t+1} = \theta_t + \alpha G_t \frac{\nabla \pi(a|s,\theta)}{\pi(a|s,\theta)}. \tag{4}$$

The complete derivation can be found in Sutton and Barto [29] and the entire algorithm is shown in Algorithm 1.

---

**Algorithm 1** REINFORCE

---

1: **Input:** differentiable policy $\pi(a|s,\theta)$.

2: Generate episode from environment (See Section 3)

3: **while** True **do** ▷ Loop until some convergence criteria is met.

4:　　Generate a sample, $S_0, A_0, R_0, \dots, S_{T-1}, A_{T-1}, R_{T-1}, S_T$ from $\pi(a|s,\theta)$

5:　　**for** $t = 0, 1, \dots, T$ **do**

6:　　　　$G_t \leftarrow \sum_{k=t+1}^{T} R_k$

7:　　　　$\theta_{t+1} \leftarrow \theta_t + \alpha G_t \nabla \ln \pi(A_t|S_t,\theta)$.

8: **Return:** optimized policy $\pi(a|s,\theta)$.

---

The REINFORCE algorithm has been well studied and a number of improvements and suggestions have been proposed [45,46,49]. The current state-of-the-art in model free policy gradient algorithms is Trust-Region Policy Optimization by Schulman et al. [45] and a simplified version of the same algorithm called Proximal Policy Optimization [46]. In this work, we restrict our attention to the former.

Trust-region policy optimization (TRPO) is an algorithm that is based on the fact that if the policy gradient update is constrained by the total variation divergence, $D_{TV}(\pi_1, \pi_2) = \max_{s \in \mathcal{S}} |\pi_1(\cdot|s) - \pi_2(\cdot|s)|$, between the old policy and the new policy, the performance of the policy is guaranteed to increase monotonically [45]. Rewriting the total variation divergence using the Kullback-Leibler divergence and introducing approximations using importance sampling, the trust-region policy optimization reduces to solving the following optimization problem:

$$
\begin{aligned}
\underset{\theta}{\text{maximize}} \quad & \mathbb{E}_{s,a \sim \pi_{\theta_{old}}} \left[ \frac{\pi_{\theta(a|s)}}{\pi_{\theta_{old}(a|s)}} q_{\theta_{old}(s,a)} \right] \\
\text{subject to} \quad & \mathbb{E}_{s,a \sim \pi_{\theta_{old}}} \left[ D_{KL}(\pi_{\theta_{old}}, \pi_\theta) \right] \leq \delta.
\end{aligned}
\tag{5}
$$

where $q_{\theta_{old}}(s,a)$ is the action-value function, i.e., the value of taking action $a$ in state $s$ when following the policy $\pi_{\theta_{old}}(s,a)$, $D_{KL}$ is the Kullback–Leibler divergence, and $\delta$ is the bound on Kullback–Leibler divergence. See Schulman et al. [45] for a complete description of the algorithm.

*2.3. Parameterized Policies*

The most common way to generate a parametric policy in a continuous action space is to use the Gaussian density function:

$$\pi(a|s,\theta) = \frac{1}{\sigma(s,\theta)\sqrt{2\pi}} \exp\left( -\frac{(a - \mu(s,\theta))^2}{2\sigma(s,\theta)^2} \right). \tag{6}$$

where $\mu(s,\theta)$ and $\sigma(s,\theta)$ are both state dependent parametric feature extractors. We use neural network feature extractors for both $\mu$ and $\sigma$ in this work. In this way, $\mu = nn_\mu(s,\theta)$ is a multilayer perceptron with three hidden layers with 100, 50, and 25 hidden neurons, respectively, where $\theta$ are the weights of the neural network, mapping the state space into the mean of the Gaussian function. We decided

to use this neural network architecture following [43], where a feedforward neural network policy with the same number of layers and hidden neurons is used to test and evaluate several tasks with continuous action spaces. $\sigma$ can either be a fixed vector, $\sigma = r \in \mathbb{R}^d$, where $d$ is the dimension of the state space, or the output of a different neural network, $\sigma = nn_\sigma(s, \theta)$. In this case, the multilayer perception used for $\sigma$ consists of two hidden layers, each with 32 hidden neurons. It is common to take the exponential of $\sigma$ to ensure a positive standard deviation [29,45]. In the multivariate case, a diagonal covariance matrix is used. For both neural networks, $\mu$ and $\sigma$, we used a non-linear *tanh* intermediate-layer activation functions, while linear activation functions are used in the output layers. Thus, the action is a sample from $\mathcal{N}(\mu, \sigma^2)$. An illustration is shown in Figure 2.

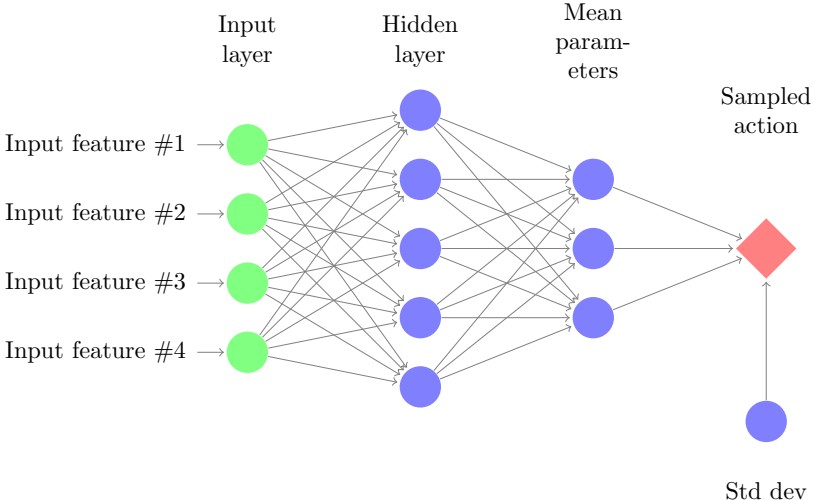

**Figure 2.** Neural network policy parameterization. The neural net maps the state, in this case a 4 dimensional space, to the mean, $\mu$, of the Gaussian policy. The output is then a sample from the Gaussian policy $\mathcal{N}(\mu, \sigma^2)$. The $\sigma$ parameter is in this work the output from a neural network.

### *2.4. Model Predictive Control*

Model predictive control (MPC) is currently the state-of-the-art for artificial pancreas systems [50–52], and is used in commercial systems including the recently FDA approved Control-IQ$^{\text{TM}}$advanced hybrid closed loop technology [53]. In general, MPC is a collection of algorithms where a model of the process is used to predict the system's future behavior. Optimal actions are then computed, while using an objective function, to ensure that the predicted behavior matches the optimal desired behaviour [54]. Algorithms typically differ in the type of model and the objective function used [54]. MPC has the advantage of incorporating constraints in the objective function. This is particularly beneficial for the artificial pancreas system case that is characterized by long delay times [54]. Because the quality of the results is completely driven by the ability of the model to describe the true process state, most MPC algorithms are used in conjunction with state estimation techniques, such as the Kalman filter [24]. The main drawback of MPC is that adapting the model to each patient individually and accounting for intra-day variability is completely dependent on the structure of the predictive model [25]. If the model is not expressive enough to capture the situation, the algorithm will fail. Several recent works have tried to lessen this burden by using multiple predictive models [55–57].

### 3. In-Silico Simulation

Most in-silico T1DM research is centered around three physiological models: the Bergman (minimal) model [58], the Hovorka model [22] and the UVA/Padova model [59], see also [60]. The minimal model is a simplified model consisting of two equations describing the internal dynamics of glucose and insulin and does not account for the significant delay involved in subcutaneous

insulin infusion. The Hovorka model and the UVA/Padova both include this delay as well as the delay in the subcutaneous glucose measurement. In this work, we use the Hovorka model.

### 3.1. Simulator

The Hovorka model consists of five compartments that describe the dynamics of glucose kinetics and insulin action [61], two external, and three internal compartments. The three internal compartments describe insulin action, glucose kinetics and glucose absorption from the gastrointestinal tract. The two external compartments describe subcutaneous insulin absorption and interstitial glucose kinetics. The original model includes one virtual patient, which we use in our experiments. In addition, we follow Boiroux et al. [24] and use model equations, parameters and distributions as given in Hovorka et al. [22] and Wilinska et al. [62] to simulate further virtual patients. Unconstrained sampling from these distributions can lead to unrealistic virtual patients, as was also pointed out in Boiroux et al. [24]. To cope with this, the samples were constrained to the following set of rules [63].

- Patient weight is sampled from a uniform distribution between 55–95 kg.
- When the basal rate is delivered and the patient is in fasting conditions, glucose levels are constant and are between 110–180 mg/dL.
- The patient's basal rates were sampled from a uniform distribution between 0.2–2.5 U.
- The patient's carbohydrate ratios were sampled from a uniform distribution between 3–30 g/U.
- Each patient is characterized with a unique insulin sensitivity factor (*ISF*) $S_i$ mg/dL/U, i.e., if an insulin bolus of size 1 U is delivered, glucose levels will drop by *ISF* mg/dL.
- The patient's insulin sensitivities were sampled from a uniform distribution between 0.5–6.5 mmol/L.
- A theoretical total daily dose (*TDD*) of insulin is computed assuming a daily diet of carbohydrates between 70–350 g. This value is then compared to sampled insulin sensitivity to ensure that the 1800 rule holds: $ISF = \dfrac{1800}{TDD}$.
- A theoretical total fraction of basal insulin is computed and is compared to *TDD* to ensure that the proportion of basal insulin is between 25–75% of *TDD*.
- All Hovorka's parameters, [62], are sampled using a log-normal distribution (to avoid negative values) around published parameters.

### 3.2. Reinforcement Learning, T1DM and the Artificial Pancreas

Because of the fact that applying reinforcement learning to any problem assumes an underlying Markov decision process, we need to take this into account when designing the state and action spaces for the T1DM case. There are several factors influencing whether or not we can interpret the glucose insulin dynamics as a Markov decision process, most notably the delayed action caused by the use of subcutaneous insulin infusion. Depending on the type of insulin used, the maximum effect of insulin is delayed and can last up to four hours [64]. On top of this comes the delay between the subcutaneous CGM measurements and the true blood glucose values, which is typically between 5 and 15 min. [65]. One of the fundamental properties of reinforcement learning algorithms, is the fact that they can control systems with delayed reward [66]. This implies that an action in a state can still be considered to be good even if the immediate reward from taking that action is not considered good. Furthermore, we note that, since the insulin infusion is the action taken by the RL agent, the environment will not change its state immediately because of the delayed insulin effect. Therefore, in this work we consider 30 min. time intervals as the time between each updated state from the environment. The insulin basal rate is kept constant during these 30 min. This will allow for the environment enough time to change significantly between each time step.

The final component involved is the reward function. In this work, we used two different reward functions, a symmetric Gaussian reward function and an asymmetric reward function, previously introduced in [67]. The Gaussian reward is given as:

$$r(g) = \exp\left\{-\frac{1}{2h^2}\left(g - g_{ref}\right)^2\right\},$$

where $g$ is the current blood glucose value, $h$ is a smoothing parameter, and $g_{ref}$ is the reference blood glucose value fixed at 108 mg/dL. The asymmetric reward function was, in [67], designed to reduce hypoglycemia and, at the same time, encouraging time-in-range. It is built as a piecewise smooth function and gives a strong negative reward for severe hypoglycemia, followed by an exponentially decreasing negative reward for hypoglycemic events starting at severe hypoglycemia, and zero reward when hyperglycemia occurs. Positive rewards from a symmetric linear function are given for glucose values in normoglycemic range. Concretely, the function is given as:

$$r(g) = \begin{cases} -100 & : g < g_{hypo^-} \\ \exp\left(\frac{\log(140.9)}{g_{hypo}}g\right) - 140.9 & : g \in [g_{hypo^-}, g_{hypo}] \\ \frac{1}{36}g - 2 & : g \in [g_{hypo}, g_{ref}] \\ -\frac{1}{72}g + \frac{5}{2} & : g \in [g_{ref}, g_{hyper}] \\ 0 & : g > g_{hyper}, \end{cases}$$

where hyperglycemia is defined as values above $g_{hyper} = 180$ mg/dL, hypoglycemia as values below $g_{hypo} = 72$ mg/dL and severe hypoglycemia as values below $g_{hypo^-} = 54$ mg/dL. Thus, the normoglycemic range are values between $[g_{hypo}, g_{hyper}]$ mg/dL. The parameters of the reward were found experimentally. Figure 3 shows a graphical representation of the reward function.

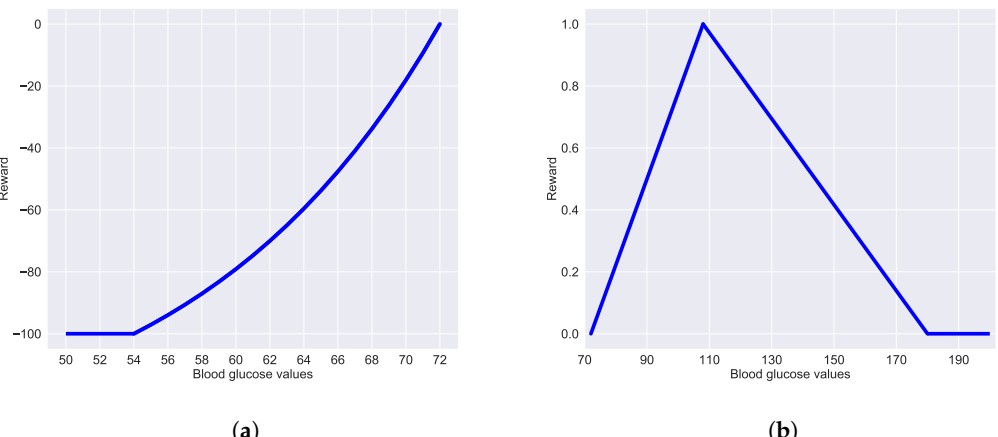

(**a**)          (**b**)

**Figure 3.** The asymmetric reward function. Low blood glucose levels (**a**) are more penalized than (**b**) high blood glucose levels.

### 3.3. Experiment Setup

The reinforcement learning agent controls the basal insulin rate of the pump which is updated every 30 min. In this work, we use two different action spaces during our experiments. Following Boiroux et al. [24], we define the action space of the agent ranging from zero, where the controller stops the insulin pump, to twice the optimal basal insulin rate (designated as TRPO in the results section). In addition, we use an extended version of the action space, which ranges from zero to three times the optimal basal insulin rate (designated as TRPOe in the results section). It is further assumed that the patient estimates and manually announces the amount of carbohydrate taken at each meal, and a bolus is given according to the patient's individual carbohydrate to insulin ratio (The number of carbohydrates that one unit of insulin will counteract). The state space of the RL agent is defined

as the concatenation of the last 30 min. of blood glucose values, the last 2 h of insulin values (last 4 actions/basal rates given in 30 min. intervals), the insulin bolus on board, which is the calculation of how much insulin is still active in the patient's body from previous bolus doses, and the size of the last given bolus if a bolus was given during the last 30 min. The insulin on board was calculated while using a decay exponential model as described in Loop (https://github.com/LoopKit/Loop). We note that using the previous insulin taken is violating the MDP assumption. We chose to keep this compromise for two reasons: (1) the insulin and carbohydrate dynamics operate on fundamentally different time scales,see e.g., [62] and (2) information about previous insulin and insulin on board is essential knowledge that the agent cannot do without.

We use time-in-range (TIR) and time-in-hypoglycemia (TIH) as the performance measures, where we want to maximize the former and minimize the latter, in order to measure the performance of our simulations. We consider the normoglycemic range as values between 72–180 mg/dL and hypoglycemia as values below 72 mg/dL, see Danne et al. [68] for further details (We ended up using 72 mg/dL as the threshold instead of 70 due to converting from the local standard of using 4 mmol/L as the hypoglycemia threshold). In addition we use the Coefficient of Variation (CoV), $\sigma/\mu$, to measure glycemic variability [69], defined as the ratio of the standard deviation to the mean of the blood glucose, and the risk index (RI), including high and low blood glucose risk indices, as described in Clarke and Kovatchev [70]. The RI measures the overall glucose variability and risks of hyper- and hypoglycemic events, while the high and low blood glucose indices (HBGI and LBGI) measure the frequency and extent of high and low blood glucose readings, respectively.

To train the algorithms, we use a standard reinforcement learning setup: (1) the agent collects episodes from the environment, followed by (2) the agent updates its parameters based on the rewards ((4) and (5)) and the process repeats until training is done (e.g., when the policy stops improving or stops changing) or the maximum number of iterations is reached. Inspired by the experiments in the original TRPO work [45], where between 50 to 200 iterations was use, we fix the number of policy update iterations to 100. This was also empirically found to provide convergence for the policies that are involved in most experiments. Furthermore, each episode is defined as starting at 00:00 and ending the next day at 12:00, at a total of 36 h. For each episode during training,s the virtual patient is given meals from a fixed-seed random meal generator to ensure that each agent is trained on the same data set. This meal generator creates four virtual meals at $\pm$ 30 min. of 08:00, 12:00, 18:00, and 22:00 h with 40, 80, 60, and 30 g of carbohydrates. $\mathcal{U}[-20, 20]$ uniform noise is added to simulate meal variation. For simplicity, the meal times are kept concurrent to the start times of each state–every 30 min. Because of the delayed meal response and the generally high variation in the bg curve, we assume that this will generalize well to meals that are taken within a state-space time interval.

To test the agents, we use a fixed set of 100 episodes with 100 daily meals scenarios, sampled from the meal generator with a different seed than the training meals. Finally, to simulate carbohydrate counting errors, all meals—both training and testing—have a counting error of $\pm 30\%$ of the exact carbohydrate count. The reinforcement learning agent was implemented using the open source reinforcement learning toolbox *garage* https://github.com/rlworkgroup/garage. [43]. The in-silico simulator was wrapped in the OpenAI Gym framework for simplified testing [71].

## 4. Results

We now present the results and discuss the performance of a simulated artificial pancreas running the TRPO algorithm described in Section 2.2 in-silico. We show the results on the original Hovorka simulated patient, [22,62], as well as a cohort of 100 simulated patients according to the parameter distributions, as given in Wilinska et al. [62]. To illustrate its potential, we compare its performance to standard basal-bolus strategy and model predictive control algorithm, as described in [6]. We begin by comparing the TRPO agent to a simple basal-bolus treatment strategy on the original Hovorka patient.

### 4.1. TRPO versus Open Loop Basal-Bolus Treatment–Hovorka Patient and Carbohydrate Counting Errors

In this simulation, we consider open loop basal-bolus therapy, i.e., a fixed optimal basal insulin rate with manually administered meal-time bolus insulin, where the optimal basal rate is calculated as the minimum amount of insulin that is required to manage normal daily blood glucose fluctuations for this particular patient, while keeping the patient at target blood glucose value during steady state. We compared the basal-bolus therapy with a hybrid closed loop system in which the TRPO agent is controlling the basal insulin rate while meal-time bolus insulin are manually administered, both strategies running the same 100 test meal scenarios. Figure 4 shows the two previously mentioned treatments superimposed over each other, where we can see the blood glucose levels for the 100 test meal scenarios. The average blood glucose values for TRPO and basal-bolus strategies are highlighted in dashed and continuous curves, respectively. The dark gray shaded area shows the maximum and minimum values for each individual step of the simulation for the TRPO agent, while the light gray shaded area does likewise with the basal-bolus regimen. We are using the maximum and minimum blood glucose values instead of a confidence interval to include all possible curves in the envelope. This is due to the severe clinical implications of even a single blood glucose curve going too low.

We see in Figure 4 that the baseline performance of the basal-bolus controller is quite good, with a high portion of time being spent within range. Still, there are several hypoglycemic events, especially after the second meal, and the variation is high, as seen in the point-wise maximum and minimum band.

In the case of the TRPO controller, we see that the hypoglycemic events after meals and the overall variance have been reduced.

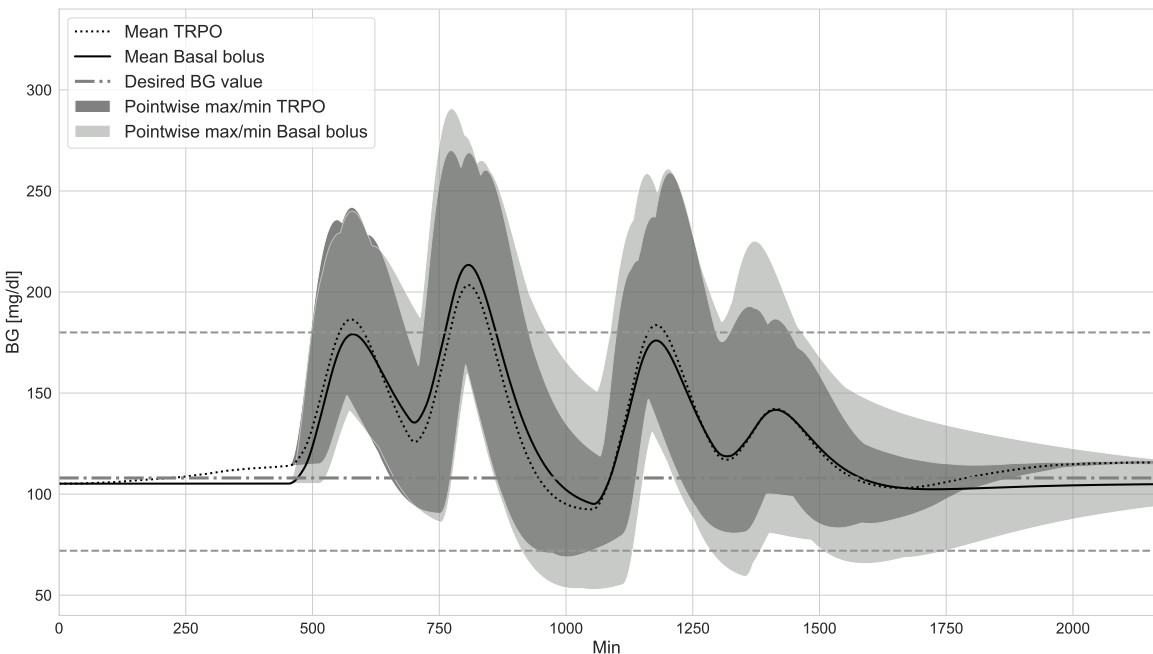

**Figure 4.** Blood glucose levels of Trust-region policy optimization (TRPO) reinforcement learning (RL) agent and standard basal-bolus therapy for the Hovorka patient. The dashed and continues curves represent the average blood glucose values over 100 test meal scenarios for the TRPO agent and the basal-bolus regimen, respectively. The shaded dark and light gray envelopes represent the minute-wise maximum and minimum blood glucose level of the simulation for the TRPO agent and the basal-bolus regimen respectively. Each test episode runs for one and a half day, a total of 2160 min.

When comparing the two results, we see that the TRPO agent has improved the results; reducing variance in general and showing better overall within range performance. Especially with respect to hypoglycemia and the glucose levels after the second meal. The TRPO agent is able to get

back to the optimal blood glucose level much quicker and with less variation than the basal-bolus strategy. An interesting observation from Figure 4 is that we see how the TRPO agent chooses to keep the steady state blood glucose value slightly higher than the desired value of 108 mg/dL (this can be observed from approximately minute 250 to 500 and from min. 1700 and onward). This helps to avoid the hypoglycemic incident that often happens after the second meal during the basal-bolus regimen.

The max-min envelope of Figure 4 is not showing the full picture with respect to the standard deviation of the two treatment options. To further illustrate this, we include kernel density plots in Figure 5, showing the distribution of the blood glucose shortly after meals, between meals and during the steady state long after any meals (equivalent to nighttime). The kernel density estimate was calculated using the *seaborn* python package (https://seaborn.pydata.org/).

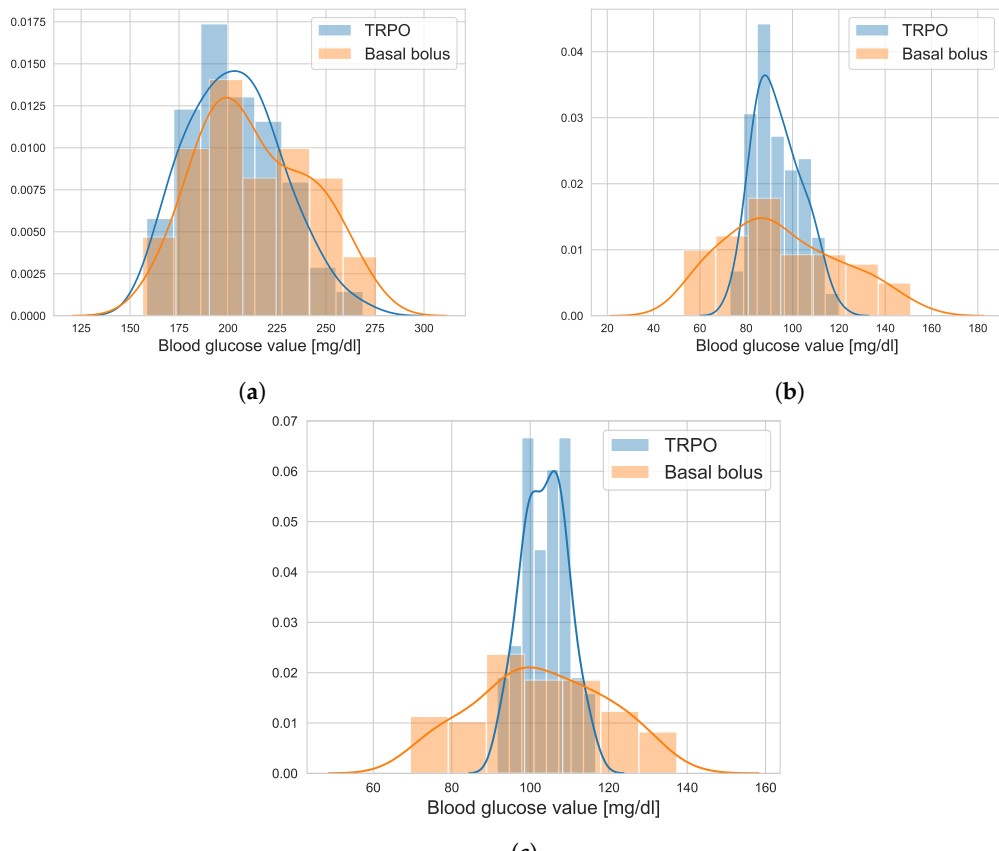

**Figure 5.** Kernel density estimation plot comparing the distribution of the results when comparing basal-bolus control to the TRPO agent. (**a**) shortly after a large carbohydrate intake, (**b**) between meals, (**c**) during nighttime (close to steady state).

It is clear, especially between meals and during night-time, that the TRPO agent treatment is superior to the basal-bolus strategy in this case.

We test the performance of the treatments in the case where the patient forgets to take the bolus insulin during a meal in order to conclude the comparison of the TRPO agent and the basal-bolus controller. To simulate this, we keep the 100 test meal scenarios, but let each meal during testing have a 0.1 probability of containing a skipped bolus. Table 1 shows a summary of all the performance measures for the experiments with the random skipped boluses (RSB) for both TRPO and basal-bolus treatments. We also include additional experiments, as shown in the Table 1, where the agent, denoted now as TRPOe, was trained with an extended action space (from zero to three times the optimal basal rate) and tested on the RSB scenarios as well as the ordinary 100 test scenarios.

Observing the Table 1, we again see that the TRPO agent is superior to the basal-bolus treatment, increasing time-in-range while decreasing time spent in hypoglycemia. It has lower variation and risk indices, and it is overall more robust towards skipped boluses. We note that the low LBGI for the skipped bolus experiment is most likely an artifact due to the blood glucose level being higher in general when there are skipped boluses involved. The same goes for the overall percentage of time spent in hypoglycemia.

**Table 1.** Summary of basal-bolus, TRPO and TRPOe results for the Hovorka patient. low blood glucose indices (LBGI) and high blood glucose indices (HBGI) is low and high blood glucose index respectively, RI is risk index, Std is the overall standard deviation and CoV is the coefficient of variation. All 100 test meal scenarios are included in the performance measures. A lower score is better for all measures, except time-in-range.

| Treatment | Time-in-Range | -Hypo | -Hyper | LBGI | HBGI | RI | Std | CoV |
|---|---|---|---|---|---|---|---|---|
| Basal-bolus | 83.45 | 2.42 | 14.13 | 0.87 | 4.62 | 5.5 | 40.35 | 0.3 |
| TRPO | 86.12 | **0.1** | 13.78 | 0.46 | **3.17** | **3.62** | **36.55** | **0.27** |
| TRPOe w/ 300 itrs | **86.33** | 0.49 | **13.18** | **0.42** | 4.14 | 4.56 | 36.71 | 0.28 |
| Random skipped boluses: | | | | | | | | |
| Basal-bolus | 79.59 | 2.27 | 18.13 | 0.85 | 5.8 | 6.65 | 50.35 | 0.36 |
| TRPO | 82.91 | **0.0** | 17.09 | **0.2** | 5.55 | 5.75 | 41.06 | **0.29** |
| TRPOe w/ 300 itrs | **84.68** | 0.49 | **14.84** | 0.43 | **4.68** | **5.11** | **40.36** | 0.3 |

When it comes to the results using the extended action space TRPOe, we found that the results using 100 policy gradient iterations are inferior to the other results. Therefore, we extended the number of training iterations to 300, which lead to an improvement over the original action space. The extended action space also leads to a treatment that is more robust to skipped boluses. However, the effect of increasing the number of policy gradient iterations from 100 to 300 represents a significant increase in data used for training the policy. There is a trade-off between the size of the action space and the number of training data/simulations needed.

## 4.2. Virtual Population Experiment: Undertreated Patients

The virtual population, as described in Section 3, have basal and bolus rates that are sub-optimal, keeping the patients within 110–180 mg/dL at steady state. We consider the virtual patients with high steady state glucose values as patients that are undertreated, i.e., their current treatment regimen does not give the desired blood glucose levels. We show a random sample of four patients in Figure 6 to illustrate the improvements made by letting a TRPO agent train and control each virtual patient. Each figure contains the original sub-optimal basal-bolus treatment as well as the results using TRPO agent superimposed over each other.

In all four cases, the TRPO agent improves the sub-optimal basal-bolus treatment. For virtual population patient #4, the performance of the basal-bolus is already close to optimal, but we still see a reduction in variance, especially later in the episode, during nighttime.

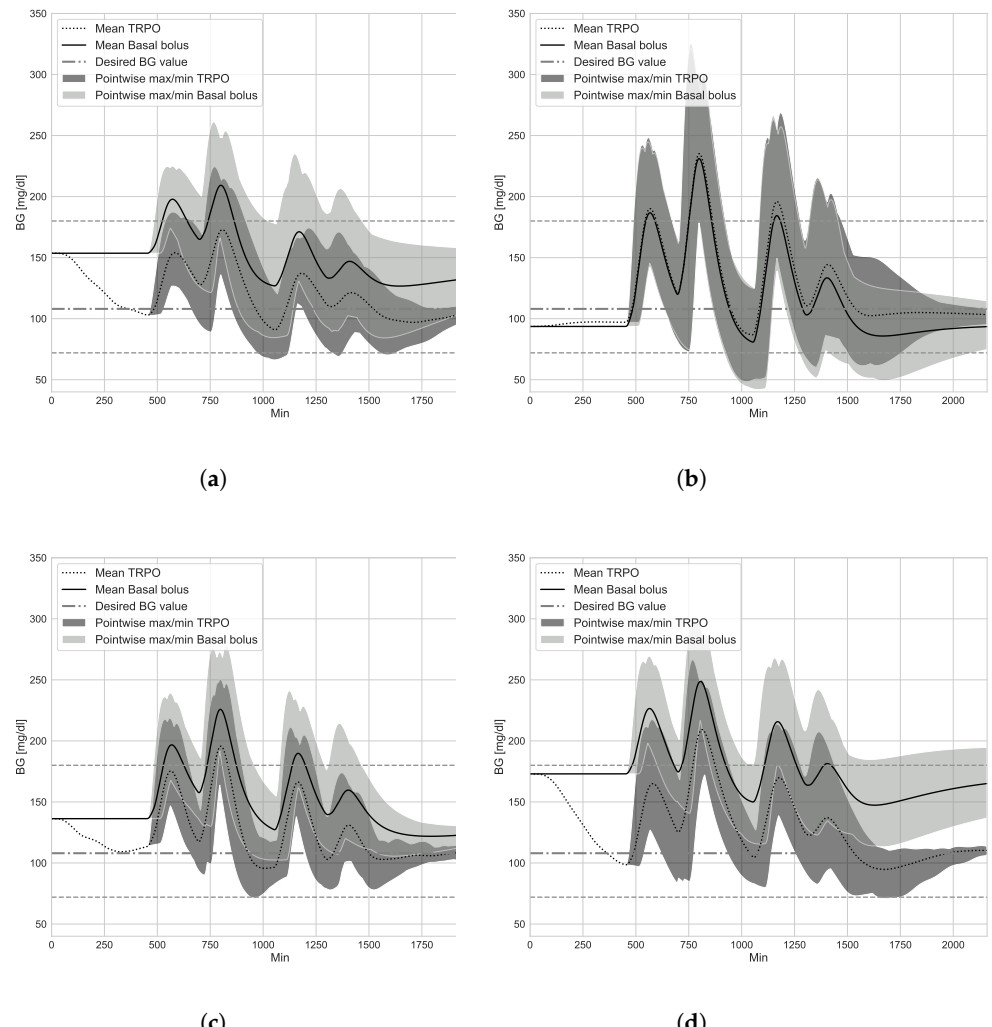

**Figure 6.** A random sample from the 100 virtual patients (**a**) patient #0, (**b**) patient #4, (**c**) patient #17, and (**d**) patient #38. All figures show results from the sub-optimal basal-bolus treatment and the TRPO agent trained on each patient individually.

*4.3. Virtual Population Experiment: TRPO versus Model Predictive Control*

We compare the TRPO agent to the open source MPC implementation (https://github.com/McGillDiabetesLab/artificial-pancreas-simulator) provided by the McGill Diabetes Lab (https://www.mcgill.ca/haidar/). The TRPO agent is individually trained for each virtual patient. The MPC controller is adapted to each patient using the total daily insulin, basal rate, and carb-ratio. As many of the patients are undertreated, some of these parameters might represent poor choices. We note that this leaves MPC at a disadvantage from the outset, since it is not able to tune the parameters during training.

In Figure 7, we see a scatterplot of the mean of the minimum and the mean of the maximum blood glucose of the 100 virtual patients controlled by MPC, the TRPO agent, and a basal-bolus strategy. This is similar to control-variability grid analysis plot [72], which is often used for measuring the quality of closed loop glucose control on a group of subjects, see e.g., [24]. The undertreated patients are left out of bounds for standard CVGA, thus requiring a different kind of analysis, as shown here.

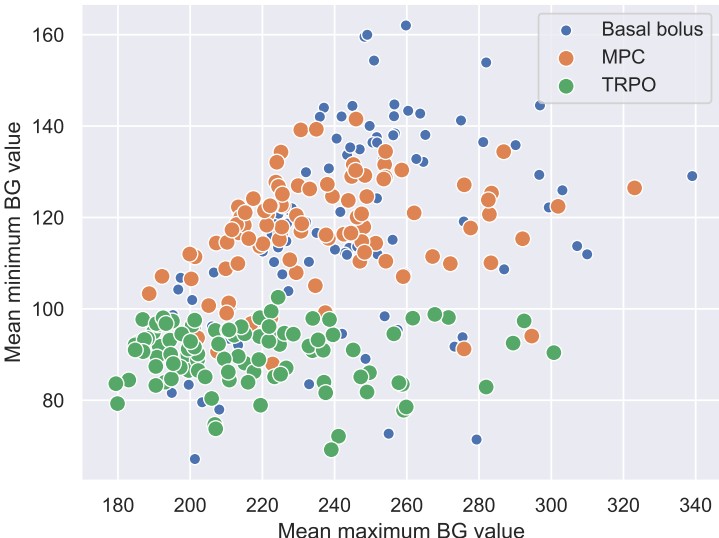

**Figure 7.** Scatterplot showing the mean of the maximum and minimum values over the 100 meal scenarios for each virtual patient on the x and y axes, respectively. MPC, basal-bolus treatment, and TRPO control is included in the plot. Tight glycemic control is in the mid to lower left area of the plot (a low maximum value and minimum value above 72 is desired).

The virtual population moves from both high mean maximum and minimum values in the basal-bolus case to lower mean maximum in MPC and even lower for the TRPO agent. We see that, in general, MPC stays at higher blood glucose levels as compared to TRPO, but conversely the TRPO agent is in some cases on the borderline low side.

To obtain a more complete picture, kernel density estimates of the same maxima and minima is shown for the entire population in Figure 8.

It is obvious that the TRPO again is outperforming the basal-bolus strategy. It shows tighter overall control and lower maximum values, while most minima are above the hypoglycemia threshold. The MPC is also tighter and improves over basal-bolus, but still the mean maximum values are, in general, higher. In addition, some of the mean minimum values are quite high, which indicates a mean blood glucose value that is generally high.

Finally, Table 2 shows the mean performance measures for the entire virtual population for basal-bolus, MPC and the TRPO agent. It also shows best and worst cases for all three treatments in terms of time-in-range (TIR) and time-in-hypo (TIH). TRPO improves the time spent in normoglycemia, while reducing the overall risk of hypo- and hyperglycemic events. However, MPC is more robust towards hypoglycemic events. Note that, in this case, in-silico patients spend less time in hypoglycemia following basal-bolus strategy than under TRPO control algorithm. This is because these patients are using sub-optimal basal-bolus treatment and therefore have higher steady state glucose values, spending most of the time close to hyperglycemia with almost no risk of hypoglycemic excursions. In this situation, the TRPO agent learns new basal rates to compensate the undertreated in-silico patients, improving the time spent in target range, but also at the same time slightly increasing the risk of hypoglycemia. Although the latter is, in general, considered to be negative, this comes down to how to design the control goals. There will always be a trade-off between better time-in-range and risk of hypo. A future study, with e.g., a parametric reward function, could help determine the exact trade-off for each patient, and take advantage of that. However, this is beyond the scope of this work.

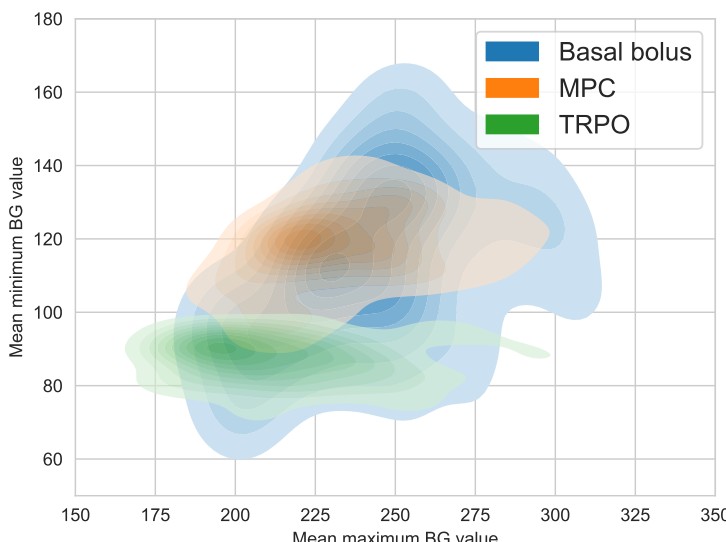

**Figure 8.** Kernel density estimate showing the approximate distribution of the same mean maxima and minima shown in Figure 7. The axes have been increased with respect to Figure 7 to fully cover the tails of the distributions.

**Table 2.** Mean performance measures for all 100 patients with all 100 test meal scenarios. Best results for each category is marked bold. For time-in-range, higher is better, for all other measures, lower is better.

| Treatment | Time-in-Range | -Hypo | LBGI | HBGI | RI | Std | CoV |
|---|---|---|---|---|---|---|---|
| Basal-bolus | 73.67 | 0.30 | 0.51 | 6.12 | 6.63 | 32.73 | 0.21 |
| TRPO | **88.72** | 0.50 | 0.78 | **3.80** | **4.57** | 32.75 | 0.24 |
| MPC | 79.25 | **0.003** | **0.13** | 5.14 | 5.27 | **30.11** | **0.19** |
| Best and worst cases: | Best TIR | Worst TIR | Worst TIH | | | | |
| Basal-bolus | 95.59 | 43.80 | 7.11 | | | | |
| TRPO | **97.18** | **63.63** | 5.01 | | | | |
| MPC | 96.02 | 55.27 | **0.15** | | | | |

## 5. Conclusions and Future Work

In this work, we have shown that policy gradient reinforcement learning using TRPO outperforms standard basal-bolus treatment and compares favourably to MPC in our experiments. We consider this work to be a strong proof of concept for the use of policy gradient algorithms in the artificial pancreas framework; the TRPO agent is able to cope with both carbohydrate counting errors and to a certain degree skipped boluses. Furthermore, the control is tighter than using a fixed optimal basal rate and risk indices are, in general, lower than both MPC and basal-bolus insulin therapy.

The main disadvantage of using RL, which has not been fully explored in this work, is the computational complexity of training. In this work, we fixed the number of policy gradient iterations to 100 for all experiments, but we empirically observed that, in many cases, far fewer iterations were required for convergence. Finally, we observed that a larger action space can lead to better control, but the increase in training data needed for convergence is significant.

All of the TRPO agents were trained model free, so from the agent's perspective the diabetes simulator is simply a black box that returns a reward when an input is given. Due to the fact that

T1DM is a well studied disease and multiple treatment strategies already exist, there is a lot of domain knowledge that gets lost in a model free setting. An obvious direction of research is including domain knowledge into the RL framework for T1DM, as in e.g., [73,74].

Finally, state-of-the-art RL contains a plethora of directions that can be explored, the perhaps most important ones for the artificial pancreas framework are inverse reinforcement learning [75], safe reinforcement learning (safe exploration) [76] and hierarchical reinforcement learning [77].

**Author Contributions:** Conceptualization, J.N.M., M.T. and I.K.L.; Funding acquisition, F.G.; Investigation, J.N.M. and M.T.; Methodology, J.N.M., M.T., I.K.L. and A.E.F.; Project administration, F.G.; Software, J.N.M., M.T. and A.E.F.; Supervision, F.G.; Validation, M.T. and A.E.F.; Writing— original draft, J.N.M., M.T. and I.K.L.; Writing—review & editing, J.N.M., M.T., I.K.L., A.E.F. and F.G. All authors have read and agreed to the published version of the manuscript.

**Funding:** J.N.M. and I.L. were funded by the Tromso Research Foundation during the course of this research.

**Conflicts of Interest:** The authors declare no conflict of interest.

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
