# Peer review of "In-Silico Evaluation of Glucose Regulation Using Policy Gradient Reinforcement Learning for Patients with Type 1 Diabetes Mellitus"

_applsci, doi:10.3390/app10186350_

Round 1

Reviewer 1 Report

In-Silico Evaluation of Glucose Regulation Using Policy Gradient Reinforcement Learning for

Patients with Type 1 Diabetes Mellitus

The paper is well written with clear structure and incremental contribution. Some comments:

  1. There are several similar papers using RL in diabetes management with adaptive insulin amount, such as

Sun, Qingnan, Marko V. Jankovic, João Budzinski, Brett Moore, Peter Diem, Christoph Stettler, and Stavroula G. Mougiakakou. "A dual mode adaptive basal-bolus advisor based on reinforcement learning." IEEE journal of biomedical and health informatics 23, no. 6 (2018): 2633-2641.

Zhu, K. Li, P. Herrero and P. Georgiou, "Basal Glucose Control in Type 1 Diabetes using Deep Reinforcement Learning: An In Silico Validation," in IEEE Journal of Biomedical and Health Informatics, doi: 10.1109/JBHI.2020.3014556.

In addition, another work

Lee, J. Kim, S. W. Park, S. Jin and S. Park, "Toward a Fully Automated Artificial Pancreas System Using a Bioinspired Reinforcement Learning Design: In Silico Validation," in IEEE Journal of Biomedical and Health Informatics, doi: 10.1109/JBHI.2020.3002022.

It uses PPO, which is a revised version of TRPO.

It would be great if the author could clarify the difference/merits of the proposed method to these works

  1. The reward function is hand designed. Why it is suitable/optimal for the algorithm’s target?
  2. The problem was assumed as a Markov decision process. The paper discusses about the delay factors. However, it seems the next state is not only depending on the last state, but also the last several states, so it might not be a typical MDP. Could you address/rebut this point a bit further?
  3. In the paper, 30 mins intervals are considered as the state space of the environment. What happens if the meal is taken within the intervals, and will it affect the performance/results?
  4. The number of iterations has been fixed at 100 policy gradient updates, which is unusual. Why?
  5. It is good to compare the proposed method to basal-bolus and MPC. Comparing to MPC, it seems TRPO has similar ‘mean maximum BG’ distribution while has smaller ‘mean minimum BG values’, concentrating between 70-100. Does it mean more hypo (which is more dangerous) exists in the experiment (also show in Table 2)? In that case, it is hard to tell which is better.
  6. Some typos in reference

Author Response

We thank the reviewer for reading the paper thoroughly and giving insightful comments. Detailed responses follow below.

Reviewer 1:
===============================================
The paper is well written with clear structure and incremental contribution. Some comments:

======= Reviewer comment =======
1. There are several similar papers using RL in diabetes management with adaptive insulin amount, such as
Sun, Qingnan, Marko V. Jankovic, João Budzinski, Brett Moore, Peter Diem, Christoph Stettler, and Stavroula G. Mougiakakou. "A dual mode adaptive basal-bolus advisor based on reinforcement learning." IEEE journal of biomedical and health informatics 23, no. 6 (2018): 2633-2641.

Zhu, K. Li, P. Herrero and P. Georgiou, "Basal Glucose Control in Type 1 Diabetes using Deep Reinforcement Learning: An In Silico Validation," in IEEE Journal of Biomedical and Health Informatics, doi: 10.1109/JBHI.2020.3014556.

In addition, another work

Lee, J. Kim, S. W. Park, S. Jin and S. Park, "Toward a Fully Automated Artificial Pancreas System Using a Bioinspired Reinforcement Learning Design: In Silico Validation," in IEEE Journal of Biomedical and Health Informatics, doi: 10.1109/JBHI.2020.3002022.

It uses PPO, which is a revised version of TRPO.

It would be great if the author could clarify the difference/merits of the proposed method to these works

======= Response =======
1. We thank the reviewer for pointing out these papers. We have included them in our introduction. We are using a different simulator as well as comparing to model predictive control.
Sun et al. learns the parameters of the insulin pump (insulin to carb-ratio) and not the insulin action itself.
Zhu et al. is quite similar to our work. They use PPO, a simpler version of TRPO, and they use the blood glucose level, bg rate and an estimate of insulin-onboard in the state space. The main difference is that they design a reward function that mimics the natural behaviour of the missing beta-cells, whereas our work focuses on a reward that encodes a more direct approach towards well-established performance measures for T1D therapy (time-in-range etc).
Lee et al. Proposes a Q learning approach, where a discrete number of actions (6) modify the given basal rate. They also operate in a dual-hormone approach, where infusion of glucagon is one of the actions. Their reward function is quite similar to ours. Finally, they provide an alternative approach to training, where a population level policy is first introduced, followed by individual adaptation to each in-silico patient.
======= Reviewer comment =======
2. The reward function is hand designed. Why it is suitable/optimal for the algorithm’s target?

======= Response =======
2. We consider the reward function to be a suitable choice in our work. It reflects clinical goals and it is piece-wise smooth such that it is in principle easier for the agent to learn (as opposed to a sparse reward, where the agent is dependent on actually reaching the optimal BG value at some point to be able to learn anything). We have changed the description of the reward function in the text. (starting on line 237)

======= Reviewer comment =======
3. The problem was assumed as a Markov decision process. The paper discusses about the delay factors. However, it seems the next state is not only depending on the last state, but also the last several states, so it might not be a typical MDP. Could you address/rebut this point a bit further?

======= Response =======
3. The problem of whether or not environments that are under RL control are true MDPs is an ongoing debate. Certainly many of the state-of-the-art results and breakthroughs in recent years, such as Atari games, are not true MDPs. Still, our motivation in this work was to at least try to approximate an MDP. By letting the actions operate on a 30 minute interval, there is a high chance that the environment has changed from one decision to the next. It is true that including insulin from previous states will violate the Markov property, but this is a compromise we choose to make since the effects the external factors (insulin and carbohydrates) have on the environment basically follows two different time scales.

======= Reviewer comment =======
4. In the paper, 30 mins intervals are considered as the state space of the environment. What happens if the meal is taken within the intervals, and will it affect the performa nce/results?

======= Response =======
4. The state space contains no information about the carbohydrate intake. If a meal is given in between state changes in the simulator, this will be reflected in a possibly increased BG during the start of the next BG state curve (recall each state is in essence a 30 minute blood glucose curve), depending on when the meal was given. It should thus not affect the results to any large degree.

======= Reviewer comment =======
5. The number of iterations has been fixed at 100 policy gradient updates, which is unusual. Why?

5. We consider this work a proof of concept for the policy gradient approach (using TRPO) for the artificial pancreas. So we have not focused extensively on hyperparameter tuning, and thus we simply made a choice of using 100 iterations as the baseline number during training.

======= Reviewer comment =======
6. It is good to compare the proposed method to basal-bolus and MPC. Comparing to MPC, it seems TRPO has similar ‘mean maximum BG’ distribution while has smaller ‘mean minimum BG values’, concentrating between 70-100. Does it mean more hypo (which is more dangerous) exists in the experiment (also show in Table 2)? In that case, it is hard to tell which is better.

======= Response =======
6. It is indeed true that the results of the TRPO agent are on average lower than the MPC, and thus has in this case a higher chance of hypoglycemia. Looking at the numbers in Table 2. we see that RL with TRPO has a significantly higher overall time in range, while the MPC is best at avoiding hypos. In the end, this comes down to how to design the control goals. There will always be a tradeoff between better time-in-range and risk of hypo. A future study, with e.g. a parametric reward function, could help determine the exact tradeoff for each patient, and take advantage of that. Again, this work should be considered more a proof-of-concept towards using model-free policy gradient RL for the artificial pancreas.

======= Reviewer comment =======
7. Some typos in reference

======= Response =======
7. We thank the reviewer for noticing this and we have done a thorough revision of the references.

Reviewer 2 Report

This manuscript introduces the concept of Policy Gradient RL approach to CGM driven insulin dosing in T1DM patients. The use of PG approach is motivated by the authors due to its ability to deal with high-dimensional continuous action spaces. 

Due to time-delayed nature of  sensor (CGM) and actuator (SubQ insulin PK) dynamics, the process of exogenous insulin delivery in T1DM patients is not fully compatible with human physiology. This is the main challenge faced by the Artificial Pancreas devices. MPC, which is the de facto state-of-the-art approach to closed-loop insulin delivery attempts to tackle this challenge by using individualized predictive models of glucose metabolism which allow to account for sensor and actuator dynamics. The authors introduce RL as an alternative to the MPC approach. However, other than describing the main features of RL, the introduction section of the manuscript does not make a clear case why RL should be used as an alternative to MPC. In section 2.4, the authors mention that the main challenge with MPC is the need for continuous model update and the risk of using insufficiently specific model. Perhaps this point should be mentioned in the introduction. 

The proposed approach uses Deep Neural Network with three hidden layers for policy representation. What were the heuristics used to establish the number of hidden neurons ? Does the number of states (observations) in this problem warrant the use of a Deep NN ? In the context of this application, the use of a DNN with large number of weights may result in the following caveat. In the training process, the agent will be exposed to certain observations more than others. Some observations may not even occur in training at all, but may occur in use, for example due to sensor failure. How do we make sure that the agent learns to perform correct actions in such cases ? Perhaps such failure conditions could be added to the simulation scenarios ?

The choice of the reward function, which is one of the key steps in the design of an RL system is based on author's previous work. It would be beneficial to the reader if one or two sentences were added to briefly describe the choice of this particular reward function. 

From the description of the experimental setup, it appears that the RL agent was trained on each virtual patient before simulation. While this is ok for a proof-of-concept demonstration, a serious question arises with regards to implementation phase. How would an RL agent be implemented in real T1DM patient if a pre-training is required ?

With regards to the comparison between RL and MPC, it appears that while the RL approach reduces hyper-glycemia, MPC is still superior in minimizing much more dangerous hypo-glycemic episodes. Given these results, this reviewer would be cautious about favorable comparison of RL w.r.t. MPC. 

Author Response

We thank the reviewer for reading the paper thoroughly and giving insightful comments. Detailed responses follows below.

This manuscript introduces the concept of Policy Gradient RL approach to CGM driven insulin dosing in T1DM patients. The use of PG approach is motivated by the authors due to its ability to deal with high-dimensional continuous action spaces.

======= Reviewer comment =======
Due to time-delayed nature of sensor (CGM) and actuator (SubQ insulin PK) dynamics, the process of exogenous insulin delivery in T1DM patients is not fully compatible with human physiology. This is the main challenge faced by the Artificial Pancreas devices. MPC, which is the de facto state-of-the-art approach to closed-loop insulin delivery attempts to tackle this challenge by using individualized predictive models of glucose metabolism which allow to account for sensor and actuator dynamics. The authors introduce RL as an alternative to the MPC approach. However, other than describing the main features of RL, the introduction section of the manuscript does not make a clear case why RL should be used as an alternative to MPC. In section 2.4, the authors mention that the main challenge with MPC is the need for continuous model update and the risk of using insufficiently specific model. Perhaps this point should be mentioned in the introduction.

======= Response =======
We agree with the reviewers comment and we have added further information to the introduction section clarifying and justifying the use of RL algorithms instead of MPC. (Line 81)

======= Reviewer comment =======
The proposed approach uses Deep Neural Network with three hidden layers for policy representation. What were the heuristics used to establish the number of hidden neurons ? Does the number of states (observations) in this problem warrant the use of a Deep NN ? In the context of this application, the use of a DNN with large number of weights may result in the following caveat. In the training process, the agent will be exposed to certain observations more than others. Some observations may not even occur in training at all, but may occur in use, for example due to sensor failure. How do we make sure that the agent learns to perform correct actions in such cases ? Perhaps such failure conditions could be added to the simulation scenarios ?

======= Response =======
We thank the reviewer for a good question and interesting comments. To start, the design of a neural network architecture given a concrete problem is still an open field of research. Furthermore, the size of the network should to some extent reflect the complexity of the problem, which in our case we believe to be quite complex. Even if the state space consists of simple insulin and blood glucose curves, the true degree of freedom and hence the complexity is quite large. In the end, we ended up using standard simple NNs similar to those used in "Benchmarking Deep Reinforcement Learning for Continuous Control" by Duan et al. from ICML 2016. In that work, several continuous action space tasks were evaluated using NN based RL. We have added an additional comment on line 163 in the paper.

Considering the second point, which can be viewed as a combination of the risk of overfitting and not enough exploration. We agree that it would make sense to add out-of-training examples as a separate experiment, but we chose to not include it in this work. We are currently working on a follow-up paper which deals with extended experiments and more realistic scenarios. We consider this work as a first stage proof of concept for policy gradient based RL for the artificial pancreas. To give further comment, the problem of exploitation vs exploration is always present when working with RL, also in our case. In real-life situations, too much exploration is not safe and perhaps not feasible, so during testing/deployment, there will always the need for additional safety layers.

======= Reviewer comment =======
The choice of the reward function, which is one of the key steps in the design of an RL system is based on author's previous work. It would be beneficial to the reader if one or two sentences were added to briefly describe the choice of this particular reward function.

======= Response =======
We agree with the reviewer's comment and we have added more to the description of the reward function.

======= Reviewer comment =======
From the description of the experimental setup, it appears that the RL agent was trained on each virtual patient before simulation. While this is ok for a proof-of-concept demonstration, a serious question arises with regards to implementation phase. How would an RL agent be implemented in real T1DM patient if a pre-training is required ?

======= Response =======
There are many answers to this question, none of which were considered in this work. For example, the RL agent might be pre-trained using historical data from the real patient before using it on the real patient itself. Another option is to use a safe RL approach in which only safe actions are taken during the training period. Finally, in "Basal Glucose Control in Type 1 Diabetes using Deep Reinforcement Learning: An In Silico Validation" by Zhu et al, they propose that an agent can be pre-trained on a virtual population average before being transferred (and fine-tuned) on the real patient.

======= Reviewer comment =======
With regards to the comparison between RL and MPC, it appears that while the RL approach reduces hyper-glycemia, MPC is still superior in minimizing much more dangerous hypo-glycemic episodes. Given these results, this reviewer would be cautious about favorable comparison of RL w.r.t. MPC.

======= Response ======== (Note, copied from rebuttal to reviewer 1 as the question is similar)
It is indeed true that the blood glucose levels of the TRPO agent are on average lower than the MPC, and thus has in this case a higher chance of hypoglycemia. Looking at the numbers in Table 2. we see that RL with TRPO has a significantly higher overall time in range, while the MPC is best at avoiding hypos. In the end, this comes down to how to design the control goals. There will always be a tradeoff between better time-in-range and risk of hypo. A future study, with e.g. a parametric reward function, could help determine the exact tradeoff for each patient, and take advantage of that. Again, this work should be considered more a proof-of-concept towards using model-free policy gradient RL for the artificial pancreas.

Round 2

Reviewer 1 Report

The answer to the reviewer only shows in the responses, but not in the main paper. All explanations are supposed to be shown in the main contexts (highlighted in the revised version as well) to clarify/address the similar questions potentially raised by readers. 

Author Response

======= Reviewer comment =======
1. There are several similar papers using RL in diabetes management with adaptive insulin amount, such as
Sun, Qingnan, Marko V. Jankovic, João Budzinski, Brett Moore, Peter Diem, Christoph Stettler, and Stavroula G. Mougiakakou. "A dual mode adaptive basal-bolus advisor based on reinforcement learning." IEEE journal of biomedical and health informatics 23, no. 6 (2018): 2633-2641.

Zhu, K. Li, P. Herrero and P. Georgiou, "Basal Glucose Control in Type 1 Diabetes using Deep Reinforcement Learning: An In Silico Validation," in IEEE Journal of Biomedical and Health Informatics, doi: 10.1109/JBHI.2020.3014556.

In addition, another work

Lee, J. Kim, S. W. Park, S. Jin and S. Park, "Toward a Fully Automated Artificial Pancreas System Using a Bioinspired Reinforcement Learning Design: In Silico Validation," in IEEE Journal of Biomedical and Health Informatics, doi: 10.1109/JBHI.2020.3002022.

It uses PPO, which is a revised version of TRPO.

It would be great if the author could clarify the difference/merits of the proposed method to these works

======= Response =======\\
1. We thank the reviewer for pointing out these papers. We have included them in our introduction in line 91, since we found them relevant to our work. Regarding the difference/merits of the proposed method, we are using a different simulator as well as comparing to the current state of the art model predictive control algorithm in the artificial pancreas systems. In addition, Sun et al. learns the parameters of the insulin pump (insulin to carb-ratio) and not the insulin action itself.
While Zhu et al. is quite similar to our work. They use PPO, a simpler version of TRPO, and they use the blood glucose level, bg rate and an estimate of insulin-onboard in the state space. The main difference is that they design a reward function that mimics the natural behaviour of the missing beta-cells, whereas our work focuses on a reward that encodes a more direct approach towards well-established performance measures for T1D therapy (time-in-range etc).
Finally, Lee et al. Proposes a Q learning approach, where a discrete number of actions (6) modify the given basal rate. They also operate in a dual-hormone approach, where infusion of glucagon is one of the actions. Their reward function is quite similar to ours. Finally, they provide an alternative approach to training, where a population level policy is first introduced, followed by individual adaptation to each in-silico patient.

We have added a section about related work under section 1.1

======= Reviewer comment =======
2. The reward function is hand designed. Why it is suitable/optimal for the algorithm’s target?

======= Response =======
2. We consider the reward function to be a suitable choice in our work. It reflects clinical goals and it is piece-wise smooth such that it is in principle easier for the agent to learn (as opposed to a sparse reward, where the agent is dependent on actually reaching the optimal BG value at some point to be able to learn anything). We have changed the description of the reward function in the text. (starting on line 237)

======= Reviewer comment =======
3. The problem was assumed as a Markov decision process. The paper discusses about the delay factors. However, it seems the next state is not only depending on the last state, but also the last several states, so it might not be a typical MDP. Could you address/rebut this point a bit further?

======= Response =======
3. The problem of whether or not environments that are under RL control are true MDPs is an ongoing debate. Certainly many of the state-of-the-art results and breakthroughs in recent years, such as Atari games, are not true MDPs. Still, our motivation in this work was to at least try to approximate an MDP. By letting the actions operate on a 30 minute interval, there is a high chance that the environment has changed from one decision to the next. It is true that including insulin from previous states will violate the Markov property, but this is a compromise we choose to make since the effects the external factors (insulin and carbohydrates) have on the environment basically follows two different time scales.
Section 3.2 (starting on line 220) has been rewritten to include the above and we have added a sentence starting on line 260.

======= Reviewer comment =======
4. In the paper, 30 mins intervals are considered as the state space of the environment. What happens if the meal is taken within the intervals, and will it affect the performance/results?

======= Response =======
4. The state space contains no information about the carbohydrate intake. If a meal is given in between state changes in the simulator, this will be reflected in a possibly increased BG during the start of the next BG state curve (recall each state is in essence a 30 minute blood glucose curve), depending on when the meal was given. It should thus not affect the results to any large degree.

======= Reviewer comment =======
5. The number of iterations has been fixed at 100 policy gradient updates, which is unusual. Why?

5. We consider this work a proof of concept for the policy gradient approach (using TRPO) for the artificial pancreas. So we have not focused extensively on hyperparameter tuning, and thus we simply made a choice of using 100 iterations as the baseline number during training.

======= Reviewer comment =======
6. It is good to compare the proposed method to basal-bolus and MPC. Comparing to MPC, it seems TRPO has similar ‘mean maximum BG’ distribution while has smaller ‘mean minimum BG values’, concentrating between 70-100. Does it mean more hypo (which is more dangerous) exists in the experiment (also show in Table 2)? In that case, it is hard to tell which is better.

======= Response =======
6. It is indeed true that the results of the TRPO agent are on average lower than the MPC, and thus has in this case a higher chance of hypoglycemia. Looking at the numbers in Table 2. we see that RL with TRPO has a significantly higher overall time in range, while the MPC is best at avoiding hypos. In the end, this comes down to how to design the control goals. There will always be a trade-off between better time-in-range and risk of hypo. A future study, with e.g. a parametric reward function, could help determine the exact trade-off for each patient, and take advantage of that. Again, this work should be considered more a proof-of-concept towards using model-free policy gradient RL for the artificial pancreas.

======= Reviewer comment =======
7. Some typos in reference

======= Response =======
7. We thank the reviewer for noticing this and we have done a thorough revision of the references.

Round 3

Reviewer 1 Report

The paper has been improved. However, there is still many typos in reference. The paper cannot be published without correcting them carefully.